# Antibody Response after 3-Dose Booster against SARS-CoV-2 mRNA Vaccine in Kidney Transplant Recipients

**DOI:** 10.3390/vaccines12030264

**Published:** 2024-03-01

**Authors:** Domenico Tripodi, Roberto Dominici, Davide Sacco, Gennaro Santorelli, Rodolfo Rivera, Sandro Acquaviva, Marino Marchisio, Paolo Brambilla, Graziana Battini, Valerio Leoni

**Affiliations:** 1Laboratory of Clinical Pathology and Toxicology, Hospital Pio Xi of Desio, Azienda Socio Sanitaria Territoriale della Brianza (ASST-Brianza), 20832 Desio, Italy; domenicotripodi96@gmail.com (D.T.); roberto.dominici@asst-brianza.it (R.D.); paolo.brambilla@unimib.it (P.B.); 2Department of Medicine and Surgery, University of Milano-Bicocca, 20126 Monza, Italy; 3Department of Brain and Behavioral Sciences, University of Pavia, 27100 Pavia, Italy; davide.sacco02@universitadipavia.it; 4Department of Molecular Genetics and Cytogenetics, Centro Diagnostico Italiano, 20147 Milan, Italy; 5Clinical Unit of Nephrology and Dialysis, Hospital Pio Xi of Desio, Azienda Socio Sanitaria Territoriale della Brianza (ASST-Brianza), 20832 Desio, Italy; gennaro.santorelli@libero.it (G.S.); rodolfo.rivera@asst-brianza.it (R.R.); graziana.battini@asst-brianza.it (G.B.); 6Diagnostics Bioprobes s.r.l. DIA.PRO, Via G. Carducci, 27, Sesto San Giovanni, 20099 Milan, Italy; sandro.acquaviva@diapro.it (S.A.); mmarchisio@diapro.it (M.M.)

**Keywords:** COVID-19, kidney transplantation, immunosuppressive drugs, S1, S2, NCP protein antigens, mRNA vaccines

## Abstract

Severe acute respiratory syndrome coronavirus 2 (SARS-CoV-2) is associated with a high rate of mortality in kidney transplant recipients (KTRs). Current vaccine strategies for KTRs seem to be unable to provide effective protection against coronavirus disease 2019 (COVID-19), and the occurrence of severe disease in some vaccinated KTRs suggested a lack of immunity. We initially analyzed the antibody response in a group of 32 kidney transplant recipients (KTRs) followed at the nephrology and dialysis unit of the Hospital Pio XI of Desio, ASST-Brianza, Italy. Thus, we studied the differences in antibody levels between subjects who contracted SARS-CoV-2 after the booster (8 individuals) and those who did not contract it (24 individuals). Furthermore, we verified if the antibody response was in any way associated with creatinine and eGFR levels. We observed a significant increase in the antibody response pre-booster compared to post-booster using both a Roche assay and DIAPRO assay. In the latter, through immunotyping, we highlight that the major contribution to this increase is specifically due to IgG S1 IgM S2. We observed a significant increase in IgA S1 and IgA NCP (*p* = 0.045, 0.02) in the subjects who contracted SARS-CoV-2. We did not find significant associations for the *p*-value corrected for false discovery rate (FDR) between the antibody response to all assays and creatinine levels. This observation allows us to confirm that patients require additional vaccine boosters due to their immunocompromised status and therapy in order to protect them from infections related to viral variants. This is in line with the data reported in the literature, and it could be worthwhile to deeply explore these phenomena to better understand the role of IgA S1 and IgA NCP antibodies in SARS-CoV-2 infection.

## 1. Introduction

The first vaccine accepted by the European Medicines Agency (EMA) was the BNT162b2 mRNA COVID-19 vaccine (Comirnaty), and the active substance in the Pfizer-BioNtech Comirnaty vaccine is the mRNA encoding the S protein of SARS-CoV-2 [1,2,3].

Kidney transplantation is the best option for patients with end-stage renal disease, and graft survival has considerably improved, mainly due to new immunosuppressive drugs to prevent rejection [4]. Immunosuppressive drugs increase the risk of infections, the most common non-cardiovascular cause of death after kidney transplantation [4]. As known, outbreak of coronavirus disease (COVID-19) led to a high morbidity and mortality in this population, who experienced severe infections because of their kidney failure and impaired immune function [4].

The most effective available mRNA vaccines against COVID-19 reach seroconversion and efficacy rates of about 95% in the general population, but in immunocompromised patients such as kidney transplant recipients (KTR), successful seroconversion ranges between 30% and 50% [4,5]. The current vaccine strategy for KTRs appears not to provide effective protection against coronavirus disease 2019 (COVID-19) [5], and the occurrence of severe COVID-19 in some vaccinated KTRs depends on lack of immunity. Vaccination induces humoral and cellular immune responses in immunocompetent subjects four or five weeks after the second dose, but kidney transplant recipients do not show seroconversion even five weeks after booster vaccination. The failed humoral response is associated with significantly lower reactive CD4+ T helper cells, the type of immunosuppressive drugs, and the type of mRNA vaccine [6]. 

The biology of SARS-CoV-2 is now well known; there are four major structural proteins in the virus: spike (S), envelope (E), membrane (M), and nucleocapsid (NCP). These are encoded by the S, E, M, and N genes, respectively. The S protein is a critical target for inducing antibodies, particularly neutralizing antibodies (Nabs): it contains the S1 and S2 subunits. This protein is docked on the surface of the virus, making it look like a “crown”, hence the name “coronavirus”. As previously reported, the S protein has two components: S1, which contains a region that is useful for binding to the target cell by adhering to the ACE2 receptor; S2, which, in a second phase, allows the virus to enter the cell.

Antibodies against all major viral antigens are detectable both during and after COVID-19, as well as after vaccination. There is a wide phenotypic variation in the human antibody response to SARS-CoV-2. Adaptive immunity involves the immunological memory and the capacity of the immune system to “learn” from many encounters with the same pathogens, thereby allowing the immune response to become more responsive and effective over time. When all three immunoglobulin classes (i.e., IgG, IgM, and IgA) are detectable, the maximum neutralization activity against SARS-CoV-2 is achieved. This is a measure of the ability of the antibodies to work together in a synergistic manner. The neutralizing antibodies (Nabs) are crucial for virus clearance and to achieve protection against the virus. They may achieve this in several ways, including interfering with virion binding to receptors, blocking virus uptake into host cells, and preventing uncoating of viral genomes in the endosome or causing aggregation of virus particles. In the case of COVID-19, however, their roles remain less defined, e.g., in terms of the predictive value of neutralization with regard to disease outcome. Neutralizing antibodies are currently the most generally recognized and accepted as truly protective against a wide range of human respiratory infections. There is hitherto no evidence of a link between in vitro neutralization titers and in vivo protection against SARS-CoV-2 [4]. The aim of this study was to evaluate the prevalence and levels of anti-SARS-CoV-2 IgG, IgA, and IgM against S1, S2, and NCP structural proteins before and after vaccination, in kidney transplant recipients with previous or no COVID-19 infection [7,8,9,10,11].

## 2. Materials and Methods

### 2.1. Patient Characteristics

Venous blood samples (3–5 mL) from all subjects were centrifuged at 1500× *g* for 15 min at room temperature and then the serum samples (CAT serum sep clot activator 3.5 mL Greiner Bio-One, Kremsmünster, Austria) were processed. The data were obtained from a group of 32 renal transplanted patients (KTRs) enrolled at the Hospital Pio XI of Desio, ASST Brianza, with a mean age ± standard deviation (SD) 63.56 ± 11.61 years, ranging from 38 to 84 years. A total of 24 males were enrolled, with an average age of 63.17 ± 10.14 years (range 38–79), and 8 females were enrolled, age 64.57 ± 16.03 years (range 39–84 years). A sample of serum was collected by venipuncture before and 17 days after the administration of the booster (3rd dose) of the mRNA vaccine BNT162b2 (Comirnaty, Pfizer-BioNTech)—to prevent coronavirus disease 2019 (COVID-19)—in order to assess the humoral immune response. Summary of the population considered is reported in Table 1.

All patients were affected by several primary pathologies responsible for chronic renal failure which required kidney transplantation and subsequent adoption of an immunosuppressive therapy regimen. In total, 16% (N = 5) of them were affected by autosomal dominant polycystic kidney disease (ADPKD), 22% (N = 7) by glomerulonephritis, and, of the remaining part, 72% (N = 20) by other diseases (Table 1). They were divided in two subgroups: previously infected individuals (COVID-19-positive) and non-infected patients (COVID-19-negative) to dissect their antibody response directed against the antigenic nucleocapsid protein, confirmed by the nasopharyngeal swab and the molecular testing (RT-PCR). Their humoral response was analyzed by typing the immunoglobulin classes. KTR patients received the 1st cycle of vaccine (2 first doses) 9–10 months before (January–February 2021). For all the 24 COVID-negative and the 8 COVID-positive subjects, serum samples were collected immediately before the 3rd dose (booster), which was administered on the same day. A 2nd serum sample was collected after 17 days (October 2021). This time interval between the 2 blood samples was due to organization requirements of the clinicians linked to the protocol of vaccine administration. 

### 2.2. Analytical Methods

Antibodies anti-SARS-CoV-2 spike (S) protein receptor-binding domain (RBD) and the nucleocapsid protein were evaluated using two methods: ElecsysR anti-SARS-CoV-2, (ECLIA) Roche Diagnostics, chosen as reference, for quantitative determination of antibodies (including IgG) to the SARS-CoV-2 spike (S) protein receptor-binding domain (RBD) in human serum, on Cobas e602 module 8000. Samples with result >0.80 BAU/mL (binding arbitrary units) were classified as positive for anti-SARS-CoV-2 antibodies, according to the manufacturer’s directions. In order to obtain a more accurate quantification, a 1:10 dilution was performed in cases of >250 BAU/mL, while a 1:100 dilution was performed in cases of >2500 BAU/mL. The other method used was the heterogeneous competitive immunoenzymatic method ACE2-RBD neutralization assay (Diagnostics bioprobes srl DIA.PRO, Italy) for the semiquantitative determination of inhibition activity of RBD-ACE2 binding induced by antibodies to SARS-CoV-2. The heterogeneous competitive immunoenzymatic method of Diagnostics bioprobes srl (DIA.PRO) measures the neutralizing activity of anti-SARS-CoV-2 antibodies in serum samples by incubating the sample to be analyzed with the spike/RBD protein. After washing, free RBD/spikes are determined by adding the recombinant ACE2 biotinylated protein and then streptavidin conjugated with horseradish peroxidase (SAV-HRP). The color is generated by tetramethylbenzidine/hydrogen peroxide (TMB/H_2_O_2_) if no antibody is bound, while a strong inhibition of color development is observed if antibodies to RBD have blocked the binding of ACE2 labeled with biotin. The presence of this antigen on the solid phase is determined by the addition of SAV-HRP, which binds to ACE2 if no neutralizing antibody is present or does not bind if the antibodies have blocked the RBD adhering to the well. The immunoglobulin classes IgA, IgG, and IgM anti-S1, S2, and NCP were analyzed before and after the 3rd booster in all the serum samples. The immunoenzymatic method allows one to carry out both the qualitative screening and the quantitative titration test. Furthermore, it is possible to carry out a typing of Ig classes produced before and after vaccination: all samples were analyzed on the basis of the isotype of immunoglobulin (Ig) classes produced (IgG, IgM, IgA) against the main antigens of SARS-CoV-2 (S1, S2, NCP) after natural infection or vaccination. Following manufacturer’s instructions, the immunotyping assay (S1, S2, NCP, IgG, IgM, IgA) provides a profile of individual antibody response in terms of % of neutralization activity, as reported in Table 2. 

The study was conducted in accordance with the Declaration of Helsinki and approved by the Comitato Etico Brianza (ABCV-Brianza 3702, March 2021). Informed consent was obtained from all subjects involved in the study.

### 2.3. Statistical Analysis

For all antibody related variables, mean, SD, median, and range (max-min) were calculated before and after the booster. To evaluate the change in antibody levels before and after the booster dose, a Wilcoxon test for paired data was employed, with significance set for *p*-value < 0.05. The patients were divided into two subgroups: those who developed the infection after the booster and those who did not develop the infection after the booster. For each patient, we calculated the delta, defined as the difference between the antibody response after the booster and the antibodies present before the booster, to evaluate the vaccine’s efficacy, considering the presence of pre-existing antibodies and to assess the differences in antibody response between the two groups. Thus, we compared the two subgroups with a Wilcoxon test for independent data, and a significant association was considered for a *p*-value < 0.05. We also evaluated whether there were differences in levels of creatinine, eGFR, age, sex, and the number of immunosuppressive therapies taken. For continuous numeric variables such as creatinine, eGFR, and age, we repeated a Wilcoxon test for independent data. Meanwhile, for factorial variables like sex and the number of therapies taken (considered as a factorial variable), a Fisher’s exact test was performed. Associations were significant for *p*-value < 0.05. To evaluate the association between the antibody levels after the booster, a linear regression model with creatinine, taking into consideration the pre-booster antibody status, age, sex, and the number of immunosuppressive therapies, was studied. The eGFR value was excluded as a regressor to avoid overcorrection of the model, as it is strongly associated with creatinine and is a value derived from age and sex. All resulting nominal *p*-values have been adjusted for the false discovery rate (FDR) to account for multiple testing, and we considered association with an adjusted *p*-value below 0.05 to be significant.

## 3. Results

As can be observed on Table 3 and Table 4, with the Wilcoxon test for paired data, we found a statistically significant difference for both the Roche assay and the DIAPRO analysis. 

As shown in the paired data boxplots in Figure 1A, there is a noticeable increase from pre-booster (T0) to post-booster (T1). 

With DIAPRO analysis through immunotyping, it was found that IgG S1, IgG S2, IgA NCP, IgM S2, and IgM NCP are significant (see Table 4 and Figure 1). However, as also observable from the boxplots in Figure 1 and the data in Table 4, IgG S2, and IgM S2 showed an increase from pre-booster to post-booster, while in the other cases, there was a slight decrease from pre-booster to post-booster. Only in the case of delta IgA S1 and IgA NCP was the observed differences between subjects significant. Additionally, there was a slightly higher delta in COVID-positive individuals compared to COVID-negative ones (see Figure 1C).

For categorical variables, such as sex and number of immunosuppressive drugs, the data are reported as counts, and the associated *p*-values are derived from the Fisher’s exact test. (Table 5). Finally, the linear regression model used to evaluate the association between post-booster antibody levels and creatinine values, while accounting for the number of immunosuppressive drugs, sex, age, and the pre-booster antibody status, did not reveal any significant associations after adjusting for the false discovery rate (FDR). However, a single significant association at the nominal *p*-value level was observed for IgM NCP, with a *p*-value of 0.024 and a regression beta coefficient of −0.34. Probably, the small size of the sample does not allow us to provide a clinical explanation for this association. The results are presented in Table 6. 

## 4. Discussion

We compared the production of all types of antibodies before and after the booster vaccine in a group of KTR patients with or without previous COVID infection. The group of COVID-positive patients, either before or after vaccination, have IgG antibodies against S1/S2/NCP. The NCP proteins of many coronaviruses are highly immunogenic and are expressed abundantly during natural infection, and high levels of IgG antibodies against NCP have been detected in sera from SARS-CoV-2 patients compared to those with no previous COVID infection. In a similar way, Blaszczuk et al. [12] found that NCP and S2 IgG antibodies were more frequently present in individuals with COVID infection. Currently approved COVID-19 mRNA vaccines generate antibodies to S1 protein, and several studies indicated that COVID-19 vaccination and SARS-CoV-2 infection induce neutralizing anti-spike antibodies and robust T-cell responses against several viral epitopes. Such responses were detectable up to one-year post-immunization, but a significant decrease was observed within the first few months. This can explain why several immunized individuals were reinfected with the virus [4,9].

Kidney transplant recipients are at high risk for fatal coronavirus disease 2019 (COVID-19), and vaccination is essential to protect this vulnerable population; unfortunately, the standard two-dose vaccination strategy has a suboptimal immunogenicity [9,12,13,14]. In a previous paper, we demonstrated that specific anti-S1 antibodies are significantly decreased 4 months post-priming dose of Comirnaty vaccine, although previous COVID-19 infection seems to intensify humoral response [15]. Further evaluation concerning antibody persistence beyond this point, and the proportion of neutralizing antibodies with higher affinity towards SARS-CoV-2, is needed, especially in naïve and immunosuppressed subjects. The investigation of humoral response to SARS-CoV-2 represents a key aspect to deal with the COVID-19 pandemic. Although neutralizing antibodies are considered to have an important protective role, the association between seropositivity and immunity, as well as the duration of protective humoral response, represent key questions of current research [13,14,16]. The FDA declared a neutralizing titer ≥1:160 as sufficient for plasma donations. However, the definition of an antibody titer conferring protection is still missing. Hall et al. [17] performed a double-blind, randomized, controlled trial of a third dose of mRNA-1273 vaccine as compared with placebo, and the primary outcome was a serologic response with an anti-receptor-binding domain (RBD) antibody level of at least 100 U/mL after the third dose, and the second outcome included the neutralization percentage. The authors concluded that the third dose of mRNA vaccine in transplant recipients had higher immunogenicity than a placebo and that the third dose booster should be considered for transplant recipients who received only two doses. Recently, a group of researchers determined the levels of protective antibodies after vaccination against COVID-19 in nearly 9000 healthcare workers, establishing for certain levels of antibody concentration, measured by two laboratory methods, increasing levels of protection. Consistent with the data in the literature, the humoral response of vaccinated KTR patients without previous infection is not optimal compared to vaccinated people with a previous natural infection. Thus, we studied the differences in antibodies levels between subjects who contracted the SARS-CoV-2 infection after the booster (8 individuals) and those who did not contract it (24 individuals). As reported in Table 3, using Wilcoxon test for paired data, we found a statistically significant difference for both the Roche assay and the DIAPRO analysis. The higher values at T1, with respect to T0, mean that the antibodies, after the booster, are produced at greater levels, which indicates the effectiveness of the vaccination stimulus, but this efficacy is partial or suboptimal for conferring high protection due to the immunosuppressive condition of the patients.

With DIAPRO analysis through immunotyping, we observed a significant increase from pre-booster to post-booster of IgG S1, IgG S2, IgA NCP, IgM S2, and IgM NCP (Table 4 and Figure 1A), while in the other cases, there was a slight decrease from pre-booster to post-booster. By analyzing delta values regarding the differences between subjects who contracted infection and those who did not, we did not find significant differences, except in the case of delta IgA S1 and IgA NCP (*p* = 0.045, *p* = 0.02), as can be observed in Figure 1C and Table 5. Additionally, in the figure with the boxplots for significant comparisons, there is a slightly higher delta in COVID-positive individuals compared to COVID-negative individuals (Figure 1C). We also verified if the antibody response was associated with the levels of creatinine and eGFR: no significant associations were observed between the antibody response to all assays and creatinine levels for the *p*-value corrected for FDR. This observation allows us to confirm that patients require additional vaccine boosters, due to their immunocompromised status and therapy, in order to protect them from infections related to viral variants. This is in line with the data reported in the literature, and it could be worthwhile to deeply explore these immunological phenomena to better understand the role of IgA S1 and IgA NCP antibodies in SARS-CoV-2 infection. Among all the subjects, a lower drug-related immunosuppression was associated with a better antibody response. After the third dose, 8/32 subjects (25%) reinfected themselves compared to only 2 subjects (6.2%) before the booster (Table 1). Given the high number of people who have survived at least one SARS-CoV-2 infection and the high vaccination coverage in the population, it was important to estimate the protective role of immunity associated with both the vaccine and the previous infection in preventing infection and severe COVID-19 disease. The maximal protection against the diagnosis of SARS-CoV-2 infection and severe disease may be achieved through hybrid immunity (the combined effect of vaccination and previous infection), while higher risk levels are always found among unvaccinated people and those without a previous diagnosis of infection. An immune response capable of determining protection and statistically correlated to it is defined as a correlate of protection (CoP), which requires identification of immunological markers and a relative threshold of protection against infection. To date, these correlates have not been unequivocally defined, although neutralizing antibodies are thought to be a critical component [17,18,19]. Our results show that kidney transplant recipients can benefit from the booster dose of COVID-19 vaccine. The only significant comparisons of delta values IgA S1 and IgA NCP have a greater value in the case of COVID-positive patients.

Limitations of our study include the relatively small number of participants, the observational, non-randomized character of the study. Kidney transplant recipients demonstrate an impaired humoral response, which also correlated with the type and number of immunosuppressive agents. Our study did not assess cell-mediated immunity. However, the data suggest that monitoring the neutralizing antibody response and total antibody concentrations, which is practically more feasible, can be used to optimize vaccination strategies evaluating the duration and degree of protection provided by vaccines. The thresholds of protection found in our study should be compared to those obtained in further studies on other populations and with a larger number of patients. It is also essential to estimate the influence of an antibody’s reduced neutralizing capacity against new emerging virus variants.

## 5. Conclusions

Our study suggests that the monitoring of the neutralizing antibody response and total antibody concentrations can be used to optimize vaccination strategies in kidney transplant recipients by evaluating the duration and degree of protection provided by vaccines. Furthermore, the immunosuppression status of the patients indicates the need for repeated vaccination stimulations (boosters) to ensure a better immune protection for this type of fragile patient; this, however, remains a difficult objective to achieve.

## Figures and Tables

**Figure 1 vaccines-12-00264-f001:**
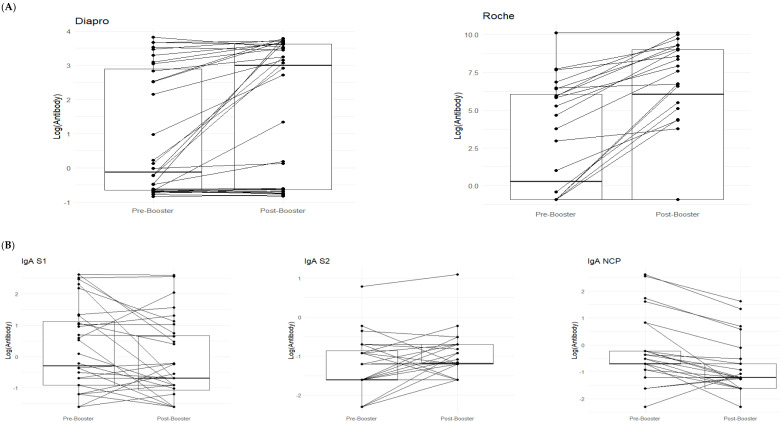
In (**A**), paired boxplots for the Roche and DIAPRO assays are displayed. In each graph, the left side represents the pre-booster status (T0) while the right side shows the post-booster scenario (T1). (**B**) shows a detailed analysis of the immunotyping assay (DIAPRO), specifically, the production of the Ig classes (IgG, IgA, IgM) targeted against the S1, S2, and NCP antigens. Results are provided as natural logarithms to enhance the graphical visualization. In (**C**), the boxplots represent delta values (post-booster—pre-booster) of the IgA S1 and IgA NCP that were significant in the Wilcoxon test, divided between those who contracted infection (colored in red) and those who did not contract it after the booster (colored in blue).

**Table 1 vaccines-12-00264-t001:** Summary of the population considered (32 KTR).

Patient	Gender	Age	Infection Pre 3° Dose	Infection Post 3° Dose	Immunosuppressive Therapy *	Creatinine (mg/dL)	eGFR (mL/min)
KTR1	F	47	NO	YES	TAC, MMF	1.48	64.4
KTR2	F	39	NO	NO	TAC, MMF, CORT	1.38	48
KTR3	F	78	NO	NO	TAC, CORT	1.63	29.9
KTR4	F	71	NO	NO	MMF, CORT, SRL	1.04	54
KTR5	F	57	NO	NO	TAC, MMF	1.16	52.2
KTR6	F	84	NO	NO	SRL	0.98	54
KTR7	F	64	NO	NO	TAC, MMF	1	59.5
KTR8	F	78	YES	YES	MMF, SRL	1.85	26
KTR9	M	63	NO	NO	TAC, MMF	1.49	49.2
KTR10	M	47	NO	NO	TAC, MMF, CORT	1.89	42
KTR11	M	69	NO	NO	AZA, CYCLO	0.82	90.2
KTR12	M	73	NO	NO	MMF, CYCLO	1.11	66
KTR13	M	75	NO	NO	MMF, CORT	1.95	33
KTR14	M	52	NO	NO	EVL	1.29	64
KTR15	M	68	NO	YES	TAC, MMF, CORT	1.12	68
KTR16	M	67	NO	NO	TAC, MMF, CORT	3.24	19
KTR17	M	59	NO	YES	TAC, MMF, CORT	1.62	46
KTR18	M	54	NO	NO	TAC, MMF	1.13	73
KTR19	M	59	NO	NO	TAC, MMF, CORT	1	82
KTR20	M	73	NO	YES	TAC, MMF, CORT	1.25	57
KTR21	M	53	NO	NO	TAC, MMF, CORT	1.07	80
KTR22	M	62	NO	NO	CORT, SRL, CYCLO	2.17	32
KTR23	M	74	NO	NO	TAC, MMF	1.32	55
KTR24	M	79	NO	NO	MMF, CORT, SRL	1.98	31
KTR25	M	38	NO	YES	TAC, MMF, CORT	1.3	70
KTR26	M	56	NO	NO	TAC, MMF, CORT	2.19	33
KTR27	M	70	NO	NO	TAC, MMF	1.52	46
KTR28	M	69	NO	YES	TAC, MMF	1.73	30
KTR29	M	76	NO	NO	CORT, EVL, CYCLO	1.49	45
KTR30	M	59	NO	NO	MMF, CYCLO	1.5	48
KTR31	M	57	NO	NO	MMF, CYCLO	2.56	25
KTR32	M	64	YES	YES	TAC, MMF	1.43	52

* Tacrolimus (TAC), N patients = 20 (63%); Corticosteroids (CORT), N = 16 (50%); Everolimus (EVL), N = 2 (6%); Mycophenolate (MMF), N = 26 (81%); Azathioprine (AZA), N = 1 (3%); Sirolimus (SRL), N = 5 (16%); Cyclosporine (CYCLO), N = 6 (19%).

**Table 2 vaccines-12-00264-t002:** Percentage of neutralization activity kindly provided by manufacturer (DIAPRO).

Follow-Up of Vaccination		
Calculate the mean OD 450 nm of NC and then the percentage of neutralization of the sample (NS%) with the following formulation NS % = 100 – [$\frac{OD\;450\;\mathrm{nm}\;Sample}{mean\;OD\;450\;\mathrm{nm}\;NC}\times 100]$		
**% of Neutralization (N_sample_%)**	**Neutralizing**	**WHO IU/mL range**
<20%	Lower or reactive	<10
20% < NS % < 30%	Moderate	10–100
30% < NS % < 60%	Good	100–400
60% < NS % < 100%	Excellent	>400

**Table 3 vaccines-12-00264-t003:** The table displays the mean, median, minimum, and maximum values for the Roche and DIAPRO assays accompanied by the respective *p*-values obtained from the Wilcoxon test for paired data; the significative values (*p* < 0.05) are reported in bold.

Diapro	Pre-Booster (T0)	Post-Booster (T1)	*p*-Value	Roche	Pre-Booster (T0)	Post-Booster T1)	*p*-Value
	(N = 32)	(N = 32)			(N = 32)	(N = 32)	
Mean (SD)	10.1 (14.3)	18.8 (17.5)	**<0.001**	Mean (SD)	2600 (7340)	5270 (8360)	**<0.001**
Median [Min, Max]	0.893 [0.430, 45.7]	20.0 [0.435, 43.8]		Median [Min, Max]	1.70 [0.400, 25,000]	486 [0.400, 25,000]	

**Table 4 vaccines-12-00264-t004:** Mean, median, minimum, and maximum values for the different IgA, IgM and IgG subtypes measured by DIAPRO assay. The Wilcoxon test for paired data is positive for *p* < 0.05.

IgA	Pre-Booster	Post-Booster	*p*-Value	IgM	Pre-Booster	Post-Booster	*p*-Value	IgG	Pre-Booster	Post-Booster	*p*-Value
	(N = 32)	(N = 32)			(N = 32)	(N = 32)			(N = 32)	(N = 32)	
**IgA S1**				**IgM_S1**				**IgG S1**			
Mean (SD)	3.14 (4.42)	2.00 (3.36)	0.119	Mean (SD)	2.70 (4.14)	1.68 (1.76)	0.742	Mean (SD)	4.35 (5.12)	6.37 (6.28)	**0.023**
Median [Min, Max]	0.750 [0.200, 13.8]	0.500 [0.200, 13.5]		Median [Min, Max]	0.400 [0.100, 13.5]	0.900 [0.100, 5.60]		Median [Min, Max]	1.50 [0.200, 13.8]	3.41 [0.200, 14.0]	
**IgA S2**				**IgM S2**				**IgG S2**			
Mean (SD)	0.375 (0.384)	0.450 (0.486)	0.076	Mean (SD)	0.366 (0.657)	0.450 (0.563)	**0.031**	Mean (SD)	1.46 (2.95)	1.25 (2.79)	**0.013**
Median [Min, Max]	0.200 [0.100, 2.20]	0.307 [0.200, 3.00]		Median [Min, Max]	0.200 [0.100, 3.80]	0.315 [0.100, 3.20]		Median [Min, Max]	0.500 [0.200, 13.2]	0.400 [0.200, 13.1]	
**IgA NCP**				**IgM NCP**				**IgG NCP**			
Mean (SD)	1.73 (3.29)	0.669 (1.09)	**<0.001**	Mean (SD)	0.963 (1.39)	0.502 (0.858)	**<0.001**	Mean (SD)	1.77 (3.09)	1.74 (3.12)	0.153
Median [Min, Max]	0.500 [0.100, 13.8]	0.300 [0.100, 5.10]		Median [Min, Max]	0.500 [0.120, 6.50]	0.300 [0.100, 4.90]		Median [Min, Max]	0.500 [0.200, 14.4]	0.500 [0.200, 14.3]	

In bold are left the subtypes of Ig and the significant *p* values.

**Table 5 vaccines-12-00264-t005:** This table presents the mean, median, maximum, and minimum values for creatinine, eGFR, age, and the delta values in antibody levels from the Roche and DIAPRO assays, categorized by COVID-negative and COVID-positive subjects. It also includes the *p*-values from the Wilcoxon test, with significant findings (*p* < 0.05) emphasized in bold.

	No COVID Infection	Yes COVID Infection	*p*-Value
	(N = 24)	(N = 8)	
Sex			
F	6 (25.0%)	2 (25.0%)	1
M	18 (75.0%)	6 (75.0%)	
Age			
Mean (SD)	65.4 (10.9)	58.1 (12.7)	0.177
Median [Min, Max]	65.5 [39.0, 84.0]	61.5 [38.0, 73.0]	
Creatinine			
Mean (SD)	1.54 (0.580)	1.48 (0.258)	0.845
Median [Min, Max]	1.44 [0.820, 3.24]	1.46 [1.12, 1.89]	
eGFR			
Mean (SD)	49.8 (19.3)	53.7 (13.9)	0.542
Median [Min, Max]	48.6 [19.0, 90.2]	54.5 [30.0, 70.0]	
N. of immunosuppressive drugs			
one	2 (8.3%)	0 (0%)	0.55
two	13 (54.2%)	3 (37.5%)	
three	9 (37.5%)	5 (62.5%)	
Delta Diapro			
Mean (SD)	10.6 (14.3)	2.77 (5.11)	0.188
Median [Min, Max]	5.99 [−11.8, 43.2]	0.0272 [−0.0485, 11.8]	
Delta_Roche			
Mean (SD)	2670 (4250)	2700 (7620)	0.088
Median [Min, Max]	218 [0, 15,600]	0 [0, 21,600]	
Delta IgG S1			
Mean (SD)	2.53 (4.23)	0.471 (1.58)	0.214
Median [Min, Max]	1.05 [−2.30, 12.1]	−0.100 [−0.200, 4.37]	
Delta IgG S2			
Mean (SD)	−0.265 (0.530)	−0.0500 (0.160)	0.239
Median [Min, Max]	−0.100 [−2.20, 0.200]	0 [−0.400, 0.100]	
Delta_IgG NCP			
Mean (SD)	−0.0467 (0.419)	0.0375 (0.200)	0.337
Median [Min, Max]	−0.0500 [−0.800, 1.40]	0 [−0.100, 0.500]	
Delta_IgM S1			
Mean (SD)	−1.20 (3.05)	−0.502 (3.46)	0.326
Median [Min, Max]	−0.0500 [−8.90, 2.30]	0.250 [−8.60, 3.40]	
Delta_IgM S2			
Mean (SD)	0.118 (0.275)	−0.0158 (0.260)	0.278
Median [Min, Max]	0.100 [−0.500, 0.700]	0.0650 [−0.600, 0.200]	
Delta_IgM NCP			
Mean (SD)	−0.465 (0.796)	−0.449 (0.676)	0.878
Median [Min, Max]	−0.250 [−3.10, 0.300]	−0.200 [−1.50, 0.227]	
Delta_IgA S1			
Mean (SD)	−1.59 (3.96)	0.208 (0.446)	**0.045**
Median [Min, Max]	−0.200 [−12.2, 5.90]	0.0500 [−0.300, 1.10]	
Delta_IgA S2			
Mean (SD)	0.0308 (0.241)	0.207 (0.286)	0.356
Median [Min, Max]	0.100 [−0.500, 0.400]	0.100 [−0.0590, 0.800]	
Delta_IgA NCP			
Mean (SD)	−1.34 (2.54)	−0.233 (0.509)	**0.022**
Median [Min, Max]	−0.300 [−10.0, 0.100]	−0.100 [−1.40, 0.245]	

**Table 6 vaccines-12-00264-t006:** Table displaying the comparison, nominal *p*-value, corrected *p*-value, and regression beta. Significant associations are highlighted in bold.

Linear Regression Model Comparison	Nominal *p*-Value	Adjusted *p*-Value (FDR)	Beta Coefficient
IgG S1~Creatinine	0.88	0.97	−0.20
IgG S2~Creatinine	0.96	0.97	−0.01
IgG NCP~Creatinine	0.49	0.97	0.09
IgM S1~Creatinine	0.77	0.97	−0.13
IgM S2~Creatinine	0.97	0.97	0.00
IgM NCP~Creatinine	**0.02**	0.27	−0.35
IgA S1~Creatinine	0.27	0.97	−1.05
IgA S2~Creatinine	0.68	0.97	−0.04
IgA NCP~Creatinine	0.31	0.97	−0.10
Roche~Creatinine	0.73	0.97	−684.87
Diapro~Creatinine	0.96	0.97	−0.22

## Data Availability

Data are contained within the article.

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
