# Peer review of "Antibody Response after 3-Dose Booster against SARS-CoV-2 mRNA Vaccine in Kidney Transplant Recipients"

_vaccines, 2024, doi:10.3390/vaccines12030264_

Round 1
Reviewer 1 Report (Previous Reviewer 4)
Comments and Suggestions for Authors
No further comments.
Author Response
We revised and improved the linguistic quality of the manuscript.
Reviewer 2 Report (Previous Reviewer 5)
Comments and Suggestions for Authors
The manuscript aimed to evaluate the prevalence and levels of anti-SARS-COV 2 IgG, IgA and IgM against S1, S2 and NCP structural proteins before and after vaccination, in kidney transplant recipients, with previous or no Covid-19 infection. The authors analyzed the immune response in a group of 32 kidney transplant recipients before and after the administration of the first booster (3° dose) of vaccine doses of the mRNA vaccine BNT162b2 (Comirnaty, Pfizer-74 BioNTech). In my opinion, the manuscript needs major revision as there some aspects that are unclear. I have few questions and comments, listed below:
· “Creatinine” does have to be written with capital letter (Abstract and lines 157, 159 etc.).
· In my opinion, the specific statistical test (Wilcoxon) as well as the level of significance (p<0.05) do not have to be given in the Abstract.
· What do the S1 and S2 stand for in the aim of the study? The authors explained that S (spike) is the structural protein of SARS-COV-2, but they did not give any background for S1 and S2.
· The centrifuging speed should be given in g. It is not possible to give it in rpm and g at the same time (line 84: “1500 g rpm”).
· Line 87: does the mean age concern all the patients or only women?
· Lines 86-91: the sentence is too long and unclear. It end with “(COVID-19 to assess the 90 humoral immune response”, which seems to be taken from another sentence.
· Table 1 is unclear. Does it consist of two parts? The table caption should be placed before the table. The legend below the Table, which looks like the second table, is unclear. I suggest giving the drugs’ names in the table, but as abbreviations (e.g. Tacrolimus as Tac). The percentages of the individual frequency of immunosuppressive drugs administered in the study group may be provided in the text. However, I suggest checking the percentage of each immunosuppressive therapy administered within the study group as according to the data mycophenolate mofetil (lack of abbreviation explanation) was administered in 26 patients (81%), but in the main Table I found number 4, which stands for MMF, in only two patients.
· Line 100: the data on the transplantation indication are not given in Table 1, therefore, the reference should not be given.
· Lines 102 and 112: the abbreviation NCP was already explained in line 60.
· I suggest dividing the Materials and Methods section to several paragraphs e.g. Patients characteristics, Analytical methods, etc., especially as the authors provided very detailed description of the analytical method.
· Line 143: if this is a table it should be numbered and the table caption should be provided.
· Line 156-161: another too long and difficult to understand sentence.
· Beginning from line 147: in the paragraph describing statistical analysis, the p value, which was considered significant should be given, instead of its appearance in the Abstract (lines 24 and 26).
· Tables 2-4 captions should be before the main Table.
· In Table 4, the numbers should contain a dot, not a comma, when specifying decimal places, hundredth places, etc.
· Line 244: there is no information in the Statistical analysis about corrected p value, which is given in Table 4. Moreover, what was the statistical test used for assessing these correlations?
Comments on the Quality of English LanguageLanguage revision is recommended as some sentences are too long and difficult to understand.
Author Response
The manuscript aimed to evaluate the prevalence and levels of anti-SARS-COV 2 IgG, IgA and IgM against S1, S2 and NCP structural proteins before and after vaccination, in kidney transplant recipients, with previous or no Covid-19 infection. The authors analysed the immune response in a group of 32 kidney transplant recipients before and after the administration of the first booster (3° dose) of vaccine doses of the mRNA vaccine BNT162b2 (Comirnaty, Pfizer-74 BioNTech). In my opinion, the manuscript needs major revision as there some aspects that are unclear. I have few questions and comments, listed below:
1) “Creatinine” does have to be written with capital letter (Abstract and lines 157, 159 etc.).
Done. “Creatinine” is written in capital letters throughout the text.
2). In my opinion, the specific statistical test (Wilcoxon) as well as the level of significance (p<0.05) do not have to be given in the Abstract.
Done. Please read the Abstract. (Lines 17-36).
3) What do the S1 and S2 stand for in the aim of the study? The authors explained that S (spike) is the structural protein of SARS-COV-2, but they did not give any background for S1 and S2.
Done. We have included a short description about S1 and S2. Lines 69-73
4) The centrifuging speed should be given in g. It is not possible to give it in rpm and g at the same time (line 84: “1500 g rpm”).
Done. See line 98.
5) Line 87: does the mean age concern all the patients or only women?
We have specified mean age of all individuals, mean age of females and mean age of males. Lines 100-104.
6) Lines 86-91: the sentence is too long and unclear. It end with “(COVID-19 to assess the 90 humoral immune response”, which seems to be taken from another sentence.
We did not find this error in the text. See lines 106-107. However, we have simplified many sentences in the text.
7) Table 1 is unclear. Does it consist of two parts? The table caption should be placed before the table. The legend below the Table, which looks like the second table, is unclear. I suggest giving the drugs’ names in the table, but as abbreviations (e.g. Tacrolimus as Tac). The percentages of the individual frequency of immunosuppressive drugs administered in the study group may be provided in the text. However, I suggest checking the percentage of each immunosuppressive therapy administered within the study group as according to the data mycophenolate mofetil (lack of abbreviation explanation) was administered in 26 patients (81%), but in the main Table I found number 4, which stands for MMF, in only two patients.
Done. Please See the Table 1.
We have placed the caption before the table 1, see line 110.
We have included the legend in table 1.
We added the drugs’ names in the table as abbreviations (e.g. Tacrolimus as Tac).
We have checked and modified the percentage of each immunosuppressive therapy.
8) Line 100: the data on the transplantation indication are not given in Table 1, therefore, the reference should not be given.
Done. We have modified the caption. See lines 107-110.
9) Lines 102 and 112: the abbreviation NCP was already explained in line 60.
Done. We have deleted the acronym “NCP”.
10) I suggest dividing the Materials and Methods section to several paragraphs e.g. Patients characteristics, Analytical methods, etc., especially as the authors provided very detailed description of the analytical method.
Done. We have divided the Materials and Methods section to several paragraphs.
11) Line 143: if this is a table it should be numbered and the table caption should be provided.
Done. We have assigned a number to the table and added a caption.
12) Line 156-161: another too long and difficult to understand sentence.
See lines 173-175. We have simplified all the long and difficult sentences.
13) Beginning from line 147: in the paragraph describing statistical analysis, the p value, which was considered significant should be given, instead of its appearance in the Abstract (lines 24 and 26).
Done. Read the statistical analysis paragraph.
14) Tables 2-4 captions should be before the main Table.
Done.
15) In Table 4, the numbers should contain a dot, not a comma, when specifying decimal places, hundredth places, etc.
We have modified the table. See table 6.
16) Line 244: there is no information in the Statistical analysis about corrected p value, which is given in Table 4. Moreover, what was the statistical test used for assessing these correlations?
We used Linear Regression Model.
We have modified the table. Please, see table 6.
Comments on the Quality of English Language
Language revision is recommended as some sentences are too long and difficult to understand.
We acknowledge the Reviewer for the positive comments.
We revised and improved the linguistic quality of the manuscript
Reviewer 3 Report (Previous Reviewer 2)
Comments and Suggestions for Authors
This study assesses the antibody response to the SARS-CoV-2 vaccine in kidney transplant recipients, a critical aspect for the transplantation community. The authors have made substantial revisions to the manuscript, and I have no additional concerns.
Author Response
We acknowledge the Reviewer for the positive comments.
We revised and improved the linguistic quality of the manuscript
Reviewer 4 Report (Previous Reviewer 3)
Comments and Suggestions for Authors
The changes suggested have been done and the present manuscript is advisable fo publication.
Author Response
We acknowledge the Reviewer for the positive comments.
We revised and improved the linguistic quality of the manuscript
Round 2
Reviewer 2 Report (Previous Reviewer 5)
Comments and Suggestions for Authors
The authors revised the manuscript and answered all my questions. However, there was an error in the first comment, it was my fault. It should be ““Creatinine” does not have to be written with capital letter (Abstract and lines 157, 159 etc.)” instead of ““Creatinine” does not have to be written with capital letter (Abstract and lines 157, 159 etc.)”.
This manuscript is a resubmission of an earlier submission. The following is a list of the peer review reports and author responses from that submission.
Round 1
Reviewer 1 Report
Comments and Suggestions for Authors
The topic of the current study is really interesting. An advantage of the current study is the evaluation of antibodies against several structural proteins.
However there are several limitations. First, the study refers to a very small number of patients. Second, the authors provide no information on participant demographics, transplantation vintage, donor type, immunosuppression, comorbidities. Third, the authors do not use any statistical method. The study is simply based on the percentatges of patients with positive antibodies. Descripitive statistics were not used and statistical significance of antibody levels between timepoints was not assessed. Furthermore, the association between antibody levels and immunosuppression burden, reported by the authors, has not been statistically examined. It is noteworthy that although the study sample is small, several correlations could be evaluated.
Finally, a point that needs clarification, in ]my opinion, is the time interval between two blooddrawns.
Methods: The time interval between two blood drawn was of 17 days (line 75).
Results: ..gap time between two bloodrawn for this group was comprised between 62 and 128 days (line 119-120)
Comments on the Quality of English LanguageThere are language issues and editing is required. [examples: Figure1 Are described humoral response (as percentage) of Ig classes against main antigenic proteins of SARS-COV-2 (line 147-148). Despite the KTR males were three times more than KTR females... (line 14)]
Reviewer 2 Report
Comments and Suggestions for Authors
Tripodi et al. investigated the immune response of kidney transplant (KT) recipients to COVID-19 vaccination. In their study, they enrolled 32 KT recipients and 23 healthy volunteers, both groups having received two doses of the vaccine 10 months prior. The study assessed antibody responses both before and after administering the third dose. The authors suggest that due to immunocompromised status, KT recipients require additional vaccine boosters.
Comment:
Kidney transplant recipients are subject to immunosuppression, a necessary measure to ensure the survival of both the graft and the patient. This immunosuppressive state significantly affects the efficacy of COVID-19 vaccinations. The importance of this study stems from its implications for the transplant community, emphasizing the need for additional vaccine boosters in KT recipients due to their immunocompromised status. The study is well-executed, offering critical insights for the transplant community.
I have one minor suggestion for improvement. The figures in the study could benefit from clearer labeling and more detailed legends. For instance, several figures currently lack a Y-axis label. Specifically, in Figures 1 and 2, it would be helpful to clearly indicate what the blue and red bars represent.
Reviewer 3 Report
Comments and Suggestions for Authors
The article by Tripodi et al. is well written and argued and is recommended for publication with minor corrections.
There are some lexical errors with the use of "don't" and several extra blanks.
The abbreviation covid19 should be explained the first time and not repeated.
Finally, the first 6 lines of the results section are actually from the materials and methods section.
Reviewer 4 Report
Comments and Suggestions for Authors
In the study conducted by Tripodi et al., the authors assessed the antibody response (including IgG titers and neutralizing antibody titers) in a cohort of 32 kidney transplant recipients who received two doses of the mRNA vaccine BNT162b2 (Comirnaty, Pfizer-BioNTech) followed by a booster. They compared the results between transplant recipients with a history of COVID-19 infection and those without, while also comparing them to healthy individuals. Addressing the following topics is crucial for a comprehensive understanding of the immunologic response in a kidney transplant setting.
Major Comments:
1) Despite the limited number of patients, incorporating statistical analyses is imperative to bolster the results. The mere presentation of antibody titers without statistical analyses hinders the ability to draw meaningful conclusions.
2) In Figures 1 and 2, clarify the significance of the blue and red bars. As currently described, it is not evident to readers, and providing this information will enhance comprehension.
3) Additional demographic data on kidney transplant recipients, such as mean age, gender, etiology of chronic kidney disease, time of transplant, immunosuppressive regimen, recent treatment for kidney allograft rejection, kidney function, and other relevant factors, would provide valuable insights.
Minor Comments:
1) In the legend of Figures 3 and 4, prioritize describing the Diapro test before the Roche test. Aligning the legend with the sequence of the graphical representation (Diapro test first, then Roche test) will facilitate clearer understanding.
Reviewer 5 Report
Comments and Suggestions for Authors
The manuscript aimed to evaluate the prevalence and levels of anti-SARS-COV 2 IgG, IgA and IgM against S1, S2 and NCP structural proteins before and after vaccination, in kidney transplant recipients, with previous or no Covid-19 infection. The authors analyzed the humoral immune response in a group of 32 kidney transplant recipients before and after the administration of the first booster (3° dose) of vaccine doses of the mRNA vaccine BNT162b2 (Comirnaty, Pfizer-74 BioNTech). They also compared the results with a group of healthy workers classified as late responders. The authors concluded that the monitoring of the neutralizing antibody response and total antibody concentrations can be used to optimize vaccination strategies in kidney transplant recipients evaluating the duration and degree of protection provided by vaccines. In my opinion, the manuscript needs major revision as there is lack of fluency and coherence between the Material and Methods, the Results and the Discussion. The obtained numerical results are difficult to follow. Also major language revision is needed as some sentences are difficult to read and to understand due to not proper grammar or stylistic errors. I have few questions and comments, listed below:
· The Abstract does not contain any results. Therefore, the conclusion given in the Abstract is not supported. What does [2] in line 14 refer to?
· In the Introduction, the authors mentioned that the current vaccine strategy for kidney transplant recipients is not effective against coronavirus disease, but they mentioned only two-dose regimen. In the Material and Methods, the authors, however, stated that the patients included in the study were before and after the administration of the first booster that is the third dose of vaccine. Are there any data on the third dose efficacy in kidney transplant recipients? I suggest adding the information about the third dose in aim of the study. Moreover, is the time interval long enough between two blood drawn (before and after the third dose) as it was only 17 days? In the Introduction, the authors stated that generally vaccination induces humoral and cellular immune response rates four or five weeks after the second dose, but in kidney transplant recipients the seroconversion was not seen even five weeks after booster vaccination. Moreover, what does gap time (62 and 128 days, median 60 days) between two blood drawn for patients and operators refer to? Please, clarify.
· The Material and Methods section needs major revision. There are no detailed characteristics of patients included in the study except for the age and sex. In the Results there is an information in the time of vaccination. In my opinion, these data should be included in Material and Methods. Moreover, the exact time of samples collection should be also given in this section. I suggest dividing this section into several paragraphs, each with specific information on patients characteristics, time of sampling, methods used for antibodies assessment and reference values assumed as positive for anti-SARS CoV-2 antibodies.
· Are there any references for two methods used for the determination of the antibodies against the SARS-CoV-2? Did the reference values follow the manufacturer’s recommendations? I suggest adding the references to make the data more reliable.
· There is no information what the red and blue bars refer to o Figures 1 and 2. Do they refer to the first and the second dose of vaccine, the study and the control group or Covid-positive and Covid-negative patients?
Comments on the Quality of English LanguageThe manuscript aimed to evaluate the prevalence and levels of anti-SARS-COV 2 IgG, IgA and IgM against S1, S2 and NCP structural proteins before and after vaccination, in kidney transplant recipients, with previous or no Covid-19 infection. The authors analyzed the humoral immune response in a group of 32 kidney transplant recipients before and after the administration of the first booster (3° dose) of vaccine doses of the mRNA vaccine BNT162b2 (Comirnaty, Pfizer-74 BioNTech). They also compared the results with a group of healthy workers classified as late responders. The authors concluded that the monitoring of the neutralizing antibody response and total antibody concentrations can be used to optimize vaccination strategies in kidney transplant recipients evaluating the duration and degree of protection provided by vaccines. In my opinion, the manuscript needs major revision as there is lack of fluency and coherence between the Material and Methods, the Results and the Discussion. The obtained numerical results are difficult to follow. Also major language revision is needed as some sentences are difficult to read and to understand due to not proper grammar or stylistic errors. I have few questions and comments, listed below:
· The Abstract does not contain any results. Therefore, the conclusion given in the Abstract is not supported. What does [2] in line 14 refer to?
· In the Introduction, the authors mentioned that the current vaccine strategy for kidney transplant recipients is not effective against coronavirus disease, but they mentioned only two-dose regimen. In the Material and Methods, the authors, however, stated that the patients included in the study were before and after the administration of the first booster that is the third dose of vaccine. Are there any data on the third dose efficacy in kidney transplant recipients? I suggest adding the information about the third dose in aim of the study. Moreover, is the time interval long enough between two blood drawn (before and after the third dose) as it was only 17 days? In the Introduction, the authors stated that generally vaccination induces humoral and cellular immune response rates four or five weeks after the second dose, but in kidney transplant recipients the seroconversion was not seen even five weeks after booster vaccination. Moreover, what does gap time (62 and 128 days, median 60 days) between two blood drawn for patients and operators refer to? Please, clarify.
· The Material and Methods section needs major revision. There are no detailed characteristics of patients included in the study except for the age and sex. In the Results there is an information in the time of vaccination. In my opinion, these data should be included in Material and Methods. Moreover, the exact time of samples collection should be also given in this section. I suggest dividing this section into several paragraphs, each with specific information on patients characteristics, time of sampling, methods used for antibodies assessment and reference values assumed as positive for anti-SARS CoV-2 antibodies.
· Are there any references for two methods used for the determination of the antibodies against the SARS-CoV-2? Did the reference values follow the manufacturer’s recommendations? I suggest adding the references to make the data more reliable.
· There is no information what the red and blue bars refer to o Figures 1 and 2. Do they refer to the first and the second dose of vaccine, the study and the control group or Covid-positive and Covid-negative patients?